REGISTERED REPORT PROTOCOL

# The effect of mother-infant group music classes on postnatal depression—A systematic review protocol

**Corinna Colella**⬚[ID]◉*, **Jenny McNeill**◉, **Fiona Lynn**◉

School of Nursing and Midwifery, Queen's University Belfast, Belfast, United Kingdom

◉ These authors contributed equally to this work.
* ccolella01@qub.ac.uk

## Abstract

**Data Availability Statement:** All relevant data from this study will be made available upon study completion.

### Background

Postnatal mental health problems affect 10–15% of women and can adversely impact on mother-infant interactions and bonding, the mother's mood, and feelings of competence. There is evidence that attending performing arts activities, such as singing, dancing, and listening to music, may improve maternal mental health with potential for an effect on postnatal depression.

### Methods

A systematic review will be conducted to assess the effect of mother-infant group music classes on postnatal depression compared to standard care, no control or wait list control. Studies will be included that report on postnatal depression. Further outcomes of interest include anxiety, stress, parenting competence, confidence and self-efficacy, perceived social support and mother-infant bonding. Infant and child outcomes measuring cognitive development, behaviour and social and emotional development will be included.

Search databases to be used will be Medline, EMBASE, CINAHL, PsycINFO, Scopus, CENTRAL, Web of Science, Maternity and Infant Care and discipline-specific journals for music.

The Cochrane's Template for Intervention description and replication (TIDieR) checklist and guide will be utilised to aid a detailed description, standardised assessment and quality assurance. Risk of bias will be assessed by the authors using the Cochrane Handbook for Systematic Reviews of Interventions risk of bias tool.

If sufficient studies are available, meta-analyses will be conducted to combine, compare and summarise the results of the studies for more precise estimates of effects. Where meta-analysis is not possible, results for each individual study will be reported through qualitative narrative data synthesis.

**Funding:** The author(s) received no specific funding for this work.

**Competing interests:** The authors have declared that no competing interests exist.

## Discussion

This systematic review will identify and synthesise evidence of the measured effect of postnatal mother-infant interventions involving music on maternal psychological and psychosocial outcomes and infant/child outcomes.

## Systematic review registration

This protocol was registered with Prospero on 18 October 2021 (registration number CRD42021283691). https://www.crd.york.ac.uk/prospero/display_record.php?ID=CRD42021283691.

## Introduction

Postnatal mental health problems, occurring after the birth of the baby, affect between 10–15 in every 100 women [1]. These problems range from mild symptoms of low mood, anxiety, difficulty coping with day-to-day living, irritability, fatigue, and loss of motivation through to postpartum psychosis and perinatal obsessive-compulsive disorder [1]. Factors such as lack of family or social support, poor housing, relationship breakdown, financial difficulty or previous traumatic life events may increase the risk of experiencing poor mental health [2]. The postnatal period is defined by the National Institute for Health and Care Excellence (NICE) as the 12 months following childbirth [3], while the World Health Organisation (WHO) recognises the first 6 weeks after childbirth as being defined as the postnatal period [4]. Postnatal mental health problems are usually identified and diagnosed by health professionals routinely involved with the mother's postnatal care including the GP, midwife or health visitor and using validated assessment tools; Edinburgh Postnatal Depression Scale (EPDS), Generalised Anxiety Disorder 7-item scale (GAD-7), Patient Health Questionnaire (PHQ-9)) [3] to score severity of symptoms. A variety of treatments are delivered to women including pharmacological (prescribed anti-depressant or anti-anxiety medication) and psychological options involving a range of modalities (cognitive behavioural therapy, counselling, psychotherapy and volunteer peer support run by local community organisations and charities). NICE guidelines, and local and national mental health policies including Regional Perinatal Mental Health Care Pathways further assist in guiding this area of practice [5].

Mental illness is also associated with several types of health inequalities between people with different demographic, socioeconomic and geographical factors with these groups of postnatal mothers more likely to be vulnerable to postnatal depression [6]. These isolated families often experience prolonged stress due to unemployment, economic hardship and social exclusion, which impact their health status negatively. In addition, they are less likely to engage with traditional parenting services. Thus, social disadvantage may create a range of difficulties for parents in terms of knowledge, skills and resources [7,8].

Postnatal mental health problems can often go unidentified, undiagnosed, and untreated for many women or they do not meet the eligibility threshold for specialist mental health services after the birth of their baby [9]. Women report a lack of identification with the concept of postnatal depression and their symptoms may not necessarily be picked up by standard assessment tools [9].

Studies investigating the effect of poor postnatal mental health have found an adverse impact on attachment [10] and bonding, self-regulation and empathy [11]. Consequently,

mothers find it difficult to engage with their infants both emotionally and behaviourally reducing their physical contact [12]. Provision by the mother of a less stimulating environment, being less attuned to their infant [13] and reduced parental competence and low mood in the mother [14] are also reported effects.

## Music based interventions and health

The arts are fluid and diverse and have traditionally been difficult conceptually to define. It has been proposed that the arts comprise of five categories; performing arts; visual arts, design and craft; literature; culture; and online, digital, and electronic arts, all of which combine active and receptive engagement and flexibility for development [15]. Arts activities often combine multiple different components and are subsequently considered complex or multimodal interventions [16]. They may involve aesthetic engagement, imagination, sensory activation, cognitive stimulation, and emotion regulation. Dependant on its nature, arts-based health interventions could include social interaction, physical activity, interaction with health care settings and engagement with themes of health, as illustrated in Fancourt's (2017) logic model linking the arts with health. Each of the individual components play their part in being linked with health outcomes. For example, emotion regulation is intrinsic to how we manage our mental health [17], given stress is a well-known risk factor for the onset and/or progression of a range of health conditions and cognitive stimulation when engaging with the arts provides the opportunity for skills development, also interrelated with mental health [18]. A key strength of arts projects is the combination of managing health promoting factors within aesthetic beauty and creative expression that provide motivation for engagement far beyond the regard of a particular aspect of good health and wellbeing [19], with further studies identifying additional benefits of resilience, vitality, purpose and quality of life [20].

Music falls under the category of performing arts [15]. Reported benefits of music include a contribution to lower levels of anxiety and biological stress in daily life [21,22] and an increase in self-esteem, confidence and self-worth [23].

The term 'music' is used within the literature to refer to a wide spectrum of activities ranging from listening to music which could be intentional or receptive on an individual basis, music listening that is shared with others, playing music using an instrument, composition of music, singing and musical movement such as dance [24]. Therefore, concise description of the music activity used within research is paramount to fully understand its benefits for participants of diverse ages, backgrounds and settings [24].

Research suggests the cause of postnatal depression has a multifactorial aetiology with biological and psychosocial risk factors [25]. The biopsychosocial model of health, first introduced by American psychiatrist, George Engel in 1977 proposed an integration of the biomedical model with psychological and social factors, which directly and indirectly impact on health [19]. This was recognised as aligning with the WHO and their 1948 definition of health as "a state of complete physical, mental, and social well-being and not merely the absence of disease or infirmity" [26]. This encompassed a biopsychosocial approach and the model is still generally recognised as the dominant theoretical model of health [19]. Psychoneuroimmunology further demonstrates the bi-directional connection between the mind and immune system [19] and the understanding of neuroplasticity further supports these vital relationships and how music can be utilised. Neuroplasticity, the connectivity and non-connectivity of neurons, networks and regions in the brain determines perception and response to stimuli in the world around us. Human behaviour is controlled by a network of neurons often with the same function and it is understood that the strength of these connections between networks can be changed [27]. Neuroplasticity does not remain the same throughout the lifespan. Primarily it

is the first 2–3 years of life, where millions of connections between neurons are being created but neuroplasticity continues at all levels throughout the lifespan of the human being [28]. Dopamine is a neurotransmitter in the brain that is shown to be present in reward-seeking behaviour, motivation [29] and reinforcement learning [30]. Neuroimaging studies have shown that listening to music may stimulate this same neural network. A neuroplasticity model of music therapy was subsequently created to provide a method using its five domains; social, emotional, cognitive, speech and communication and movement, to explain how neuroplasticity can be enhanced by music [27]. Further research seeking to explore the potential benefits of music among women with postnatal depression would therefore be timely.

A significant established psychosocial risk factor of postnatal depression is low, or lack of, social support [31]. A key ingredient of group interventions are the provision of social support and cohesion and the involvement of a synchronised activity, such as singing and dancing. Social identity theory identifies the extent to which group members form a shared social identity and determines whether being part of a group influences participants mental health [32]. Singing, due to its propensity to bond people has been established as an effective means of encouraging such identification [33]. Parent and baby groups are well placed to provide a group-based environment and are widely available in various formats involving music and singing for postnatal mothers to attend with their infant.

The purpose of reviewing the effect of mother-infant group music classes on postnatal depression through a systematic process, is to evidence the measured effect on maternal and infant outcomes. Following searches of Prospero and the Cochrane library for completed or ongoing reviews, no systematic review addressing the effect of mother-infant group music classes on postnatal depression were found and we therefore seek to bring evidence together within this systematic review to inform practice and future research within postnatal mental health. The added benefit of including qualitative studies in the review is to have the opportunity to consider existing research that has explored women's perceptions, experiences, and perspectives of attending group music classes with their infant, which will allow further understanding of their feasibility and acceptability.

### Objectives

1. Systematically search and review research evidence that assesses the effect of mother-infant group music classes on postnatal depression for women $\leq$ 12 months post-partum with an infant aged $\leq$ 12 months (at enrolment), compared to women who have received standard care, a comparative intervention, no care or included a wait list control

2. Review process evaluations and qualitative studies conducted alongside studies eligible for inclusion

3. Interpret findings to inform future research and practice

### Methods

This systematic review protocol has been developed according to the preferred Reporting Items for Systematic Reviews and Meta-Analyses Protocols (PRISMA-P) [34]. A PICOS (participants, interventions, comparators, outcomes, study design) framework was developed to identify criteria for study inclusion as follows:

### Inclusion criteria

**Participants.**  We will include studies that recruited women $\leq$ 12 months post-partum with an infant aged $\leq$ 12 months at enrolment. We will also include women who were enrolled in the antenatal period, but the intervention phase commenced in the postnatal period. We will include studies whether or not study eligibility criteria required women to be screened and meet a given threshold for outcomes of interest on recruitment, such as postnatal depression or anxiety.

**Interventions.**  We will focus on studies that consist of a mother-infant group intervention lasting between 6–12 weeks in duration, that comprises of an intervention including music only or in a multi component format where a significant proportion of the intervention includes music.

**Comparators.**  Comparators will include groups of women who have received standard care, a comparative intervention, no care or included a wait list control.

### Primary outcome

- Postnatal depression

We will include any outcome measure for postnatal depression, including but not limited to clinically validated assessment tools, such as the such as the Diagnostic and Statistical Manual of Mental Disorders, fifth edition (DSM-5), and self-reported measurement tools.

**Secondary outcomes.**  Maternal outcomes:

- Maternal stress/anxiety

- Parenting stress

- Parenting competence, confidence, or self-efficacy

- Perceived social support

- Maternal-infant interaction/bonding

   Infant/child outcomes:

- Cognitive development

- Behaviour

- Social and emotional development

### Feasibility and acceptability outcomes

- Recruitment rate

- Adherence and participation rates

- Attrition and retention rates and reasons for dropout

- Mothers' satisfaction with the intervention

### Exclusion criteria

- Women >12 months postpartum with an infant >12 months at enrolment

- Interventions that contain only one song, one episode of music listening or one episode of music making as it will not represent the majority of the intervention format

- Studies without a control group

Studies will be included that use standardised measurement tools that provide continuous or dichotomous outcome data for all maternal and infant/child outcome measures.

The timing of the first outcome assessment must be < 3 months post-intervention. There will be no limit on the timing of the final follow up to allow longer follow up durations to be included in the review. However, for the purposes of the meta-analysis, we will extract outcome data at, or nearest to, the following time points: immediately post-intervention, at 3, 6, 12, 18 and 24 months post-intervention.

We will include randomised controlled trials (RCTs) and non-randomised controlled trials. The Canada's Drug and Health Technology Agency (CADTH) search filter will be utilised which is sensitive to searching for non-randomised controlled trials, as well as RCTs.

Process evaluations and qualitative studies conducted alongside studies eligible for inclusion will also be reviewed to provide useful insight on the feasibility and acceptability of the intervention and study in terms of recruitment, adherence, attrition, retention, and satisfaction with the intervention. For example, the timing and setting of the intervention in included studies; level of adherence to the intervention by those randomly assigned to the experimental group and reasons for adherence/non-adherence.

Due to its multi-disciplinary nature, one of the many challenges within arts in health research is the many fields in which it is spread; arts, public health, medicine, wellbeing, and psychology [19]. Therefore, searches of a range of databases are required that will capture the broad spectrum of this topic. Electronic bibliographic databases will include Medline, Excerpta Medica Database (EMBASE), Cumulative Index to Nursing and Allied Health Literature (CINAHL), PsycINFO, Scopus, Cochrane Central Register of Controlled Trials (CENTRAL), Web of Science, Latin American and Caribbean Health Sciences Literature (LILACS), Maternity and Infant Care, and discipline specific journals for music. In addition, backward and forward searching of reference lists of included studies will be conducted, a search of unpublished / grey literature through Conference Proceedings Citation Index, Google Scholar, OpenGrey, ProQuest Dissertations and Theses, and clinical trial databases, such as clinicaltrials.gov, will be completed.

Medical subject headings (MeSH) were used to develop the list of search terms supplemented with text word terms to capture as wide a range of records as possible. Search terms include—Women, Mothers, Infant, Mother-infant, Postpartum period, Postnatal care, Music, Music therapy, Dancing, Randomized Controlled Trials, Random Allocation, Systematic review. An example of search terms and syntax used for MEDLINE (Table 1). Databases will be searched from inception. No restrictions will be imposed regarding language or year of publication.

We will use the Covidence online software platform for importing eligible studies, removing duplicates, screening titles and abstracts, and for full text review. Following removal of duplicates, two authors (CC and FL) will independently screen titles and abstracts for 10% of records and assess level of agreement. This process will continue in increments of 5% until at least 80% agreement is reached, after which the lead reviewer (CC) will continue to screen the remaining titles and abstracts. At full text stage, each record will be independently screened by

**Table 1. Search terms and syntax for database searches, example from MEDLINE.**

| MEDLINE® Search |
| --- |
| 1 Women/ |
| 2 Mothers/ |
| 3 Caregivers/ |
| 4 Parents/ |
| 5 maternal (ti,ab,hw,kf,kw) |
| 6 mother-infant (ti,ab,hw,kf,kw) |
| 7 OR/1-6 |
| 8 exp Infant/ |
| 9 exp Infant, Newborn/ |
| 10 baby (ti,ab,hw,kf,kw) |
| 11 babies (ti,ab,hw,kf,kw) |
| 12 toddler (ti,ab,hw,kf,kw) |
| 13 child 0–2 (ti,ab,hw,kf,kw) |
| 14 OR/8-13 |
| 15 exp Music/ |
| 16 exp Music Therapy/ |
| 17 Dancing/ |
| 18 Singing/ |
| 19 Acoustics/ |
| 20 Sound/ |
| 21 sing (ti,ab,hw,kf,kw) |
| 22 song (ti,ab,hw,kf,kw) |
| 23 rhythm (ti,ab,hw,kf,kw) |
| 24 melody (ti,ab,hw,kf,kw) |
| 25 lullaby (ti,ab,hw,kf,kw) |
| 26 OR/15-25 |
| 27 postnatal (ti,ab,hw,kf,kw) |
| 28 Postnatal Care/ |
| 29 postnatal period (ti,ab,hw,kf,kw) |
| 30 postnatal mothers (ti,ab,hw,kf,kw) |
| 31 Postpartum Period/ |
| 32 postpartum mothers (ti,ab,hw,kf,kw) |
| 33 Perinatal Care/ |
| 34 OR/27-33 |
| 35 (randomized controlled trial or controlled clinical trial or pragmatic clinical trial or equivalence trial or clinical trial, phase III).pt. |
| 36 Randomized Controlled Trial/ |
| 37 exp Randomized Controlled Trials as Topic/ |
| 38 "randomized controlled trial" (ti,ab,hw,kf,kw) |
| 39 exp Controlled Clinical Trial/ |
| 40 "controlled clinical trial" (ti,ab,hw,kf,kw) |
| 41 Random Allocation/ |
| 42 Double-Blind Method/ |
| 43 double blind procedure (ti,ab,hw,kf,kw) |
| 44 double-blind studies (ti,ab,hw,kf,kw) |
| 45 Single-Blind Method/ |
| 46 single blind procedure (ti,ab,hw,kf,kw) |
| 47 single-blind studies (ti,ab,hw,kf,kw) |
| 48 Placebos/ |
| 49 placebo (ti,ab,hw,kf,kw) |
| 50 Control Groups/ |
| 51 control group (ti,ab,hw,kf,kw) |
| 52 (random* or sham or placebo*) (ti,ab,hw,kf,kw) |
| 53 ((singl* or doubl*) adj (blind* or dumm* or mask*)) (ti,ab,hw,kf,kw) |
| 54 (Nonrandom* or non random* or non-random* or quasi -random* or quasirandom*) (ti,ab,hw,kf,kw) |
| 55 Allocated (ti,ab,hw) |
| 56 ((open label or open-label) adj5 (study or studies or trial*)) (ti,ab,hw,kf,kw) |
| 57 ((equivalence or superiority or non-inferiority) adj3 (study or studies or trial*)) (ti,ab,hw,kf,kw) |
| 58 (pragmatic study or pragmatic studies) (ti,ab,hw,kf,kw) |
| 59 ((pragmatic or practical) adj3 trial*) (ti,ab,hw,kf,kw) |
| 60 ((quasiexperimental or quasi-experimental) adj3 (study or studies or trial*)) (ti,ab,hw,kf,kw) |
| 61 (phase adj3 (III or "3") adj3 (study or studies or trial*)) (ti,ab,hw,kf,kw) |
| 62 "systematic review"/ |
| 63 OR/35-62 |
| 64 7 and 14 and 26 and 34 and 63 |

two authors. Any disagreements will be resolved by discussion and consensus of the review team. To ensure the transparent reporting of identified studies, we will include a PRISMA flow chart in the systematic review, which will illustrate the article selection process [35]. This flow chart will map out the number of studies identified, included and excluded at each stage, and reasons for exclusion at full text review.

A standardised data extraction form will be agreed prior to commencement to ensure data extraction is consistent. Data from multiple reports from the same study will be extracted using a single data extraction form. Data extraction for each study will be conducted independently by two authors (CC and FL/JM) using Covidence software. Extracted data will include the title, authors, year of publication, location of study, population (participant socio demographic characteristics; sample size—initial and final, participant level of postnatal depression, if reported), intervention type and dosage, control group type and dosage, outcomes (measures; time interval of measurement; instruments used), type of analysis, results, recruitment rates, adherence rates, attrition/retention rates, and reasons if given. Outcome data will be exported to Review Manager 5 software for analysis. Any unreported outcome data will not be requested from the study investigators due to the time constraints of the primary author's PhD studies.

To comprehensively describe each intervention analysed within the systematic review, the Cochrane's Template for Intervention Description and Replication (TIDieR) checklist and guide [36] will be utilised, this will aid a standardised assessment and quality assurance.

Risk of bias will be independently assessed by two authors using the Cochrane Handbook for Systematic Reviews of Interventions Risk of Bias tool [37] for randomised controlled trials and the Risk Of Bias In Non-randomized Studies—of Interventions (ROBINS-I) tool for non-randomised studies [38]. Each study will be evaluated by considering the six bias domains: selection, performance, detection, attrition, reporting and other. Any disagreement will be resolved by discussion and consensus reached. Studies will be classified as at low, uncertain, or high risk of bias according to the criteria and using the traffic light system, this will inform data synthesis by illustrating the overall quality of the studies.

For continuous data, the mean and standard deviation (SD) for each group and group size will be extracted and mean differences calculated. For dichotomous data, the number with each event and sample size will be extracted and odds ratios calculated.

If sufficient studies of similar interventions are available, we will use meta-analysis to combine, compare and summarise the results of the studies so more precise estimates of the effects can be made in terms of the population, intervention, comparator, and outcomes. We will prioritise RCT data for the meta-analysis. If non-randomised controlled trials are identified, they will be synthesised separately through a narrative summary. The studies used must compare the same type of intervention and measure the same outcomes. When outcomes are measured on the same scale, the mean difference (MD) will be calculated and if studies use different scales to measure the same outcome, we will calculate the standardised mean difference (SMD) and corresponding 95% CI for continuous outcomes. In addition to this, differences between duration of intervention, type of setting, group size, and type of facilitator between studies will be assessed via subgroup analysis to establish what has yielded the most significant retention and satisfaction outcomes.

Where meta-analysis is not possible, results will be reported through qualitative narrative data synthesis following the principles of thematic analysis [39]. Papers will be read closely, and an index paper will be identified that reflects the focus of the review most accurately. Then themes and findings will be coded, and an initial thematic framework will be created on a spreadsheet. All remaining papers will be coded and mapped onto this framework. This process will identify similarities and differences in emerging themes. Once the framework is

agreed by all authors (CC, FL and JM), the Confidence in the Evidence from Reviews of Qualitative research (GRADE-CERQual) framework for systematically assessing confidence in review findings will be used to establish methodological limitations, coherence, adequacy of data and relevance [40]. Consensus between two authors (CC and FL/JM) will be agreed following this process. Review findings will be graded for confidence using a classification system of high, moderate, low or very low confidence. The use of the GRADE-CERQual approach is to produce transparent judgement about confidence in qualitative evidence and facilitate the use of qualitative evidence to address a range of issues, including the acceptability and feasibility of interventions [40].

## Assessment of heterogeneity

If the number of included studies is low or has small sample sizes, statistical tests for heterogeneity may have low power and be difficult to interpret [37]. If there are sufficient studies to perform a meta-analysis, an assessment of heterogeneity will be conducted by visually examining forest plots for consistency of results and by calculating the $I^2$ statistic, which represents the percentage of effect estimate variability that is due to heterogeneity instead of sampling error [37].

## Subgroup analyses

It is intended to complete analyses of subgroups, to assess intervention effects for specific characteristics of the participants.

The following characteristics will be used for subgroup analysis;

- women were ≤6 months postpartum at enrolment

- at least 70% of women recruited met at least one baseline characteristic representative of social disadvantage; <20 years of age, ethnic minority group, low-income household, or single parent household

- women with an existing diagnosis of a mental health condition

- Women recruited were screened and met a given threshold for postnatal depression, in accordance with the inclusion criteria for the study.

## Sensitivity analysis

If sufficient studies are obtained, sensitivity analysis will be conducted to examine the impact of a high risk of bias. We will systematically remove studies with high selection, performance, attrition, detection and reporting bias. Funnel plots will be used to determine publication bias. The potential impact on the findings when assessing the risk of bias will be reported on. For example, whether the results from the meta-analysis are robust following exclusion of studies that are at high risk of attrition bias.

## Patient and public involvement

The proposed study is a systematic review of the literature, to gain insight for future studies and research. Hartbeeps, a third sector organisation established worldwide providing weekly music-based parent and infant/child interventions will be consulted for assistance with interpretation of findings in relation to the interventions used in the studies and with the components of those interventions.

## Discussion

The proposed systematic review will provide evidence of effect of music-based parent/infant interventions aimed at mothers within the postnatal period (≤12 months) and the maternal psychological and psychosocial outcomes the studies measure and report. Infant/child outcomes will also be reported. The overview of included studies will include the country of origin and similarities and differences in their national maternity policy approach to identify the context in which the population sits. It will offer information on the types of intervention currently explored within global experimental research, what components are mostly commonly used in the format of the music-based intervention, the feasibility of the intervention within the target population group in terms of recruitment, participation, retention and acceptability, and the extent and diversity of the psychological and psychosocial outcomes they measure. This understanding of the current evidence base will provide a solid foundation of knowledge to guide further experimental study.

## Dissemination

The protocol and systematic review will be completed as part of a PhD study conducted within the School of Nursing and Midwifery at Queens University, Belfast, Northern Ireland. Findings will be disseminated via academic audiences, relevant stakeholders and service users and social media.

## Supporting information

**S1 Appendix. PRISMA-P checklist.**
(DOC)

## Author Contributions

**Conceptualization:** Corinna Colella, Jenny McNeill, Fiona Lynn.

**Writing – original draft:** Corinna Colella.

**Writing – review & editing:** Jenny McNeill, Fiona Lynn.

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
