## [Decision Letter · Decision Letter 0]

2 May 2022

PONE-D-21-35789

The effect of mother-infant group music classes on postnatal depression – a systematic review protocol

PLOS ONE

Dear Dr. Colella,

Thank you for submitting your manuscript to PLOS ONE. After careful consideration, we feel that it has merit but does not fully meet PLOS ONE’s publication criteria as it currently stands. Therefore, we invite you to submit a revised version of the manuscript that addresses the points raised during the review process.

We would like you to focus especially on the comments of reviewer 1. Please be as clear as possible on the inclusion criteria for the publications, e.g. the diagnostic characteristics of the patients, randomization, and control conditions. Also, please state how potential biases might be dealt with.

We look forward to receiving your revised manuscript.

Kind regards,

Astrid M. Kamperman

Academic Editor

PLOS ONE

Reviewers' comments:

Reviewer's Responses to Questions

**Comments to the Author**

1. Does the manuscript provide a valid rationale for the proposed study, with clearly identified and justified research questions?

Reviewer #1: Yes

Reviewer #2: Yes

2. Is the protocol technically sound and planned in a manner that will lead to a meaningful outcome and allow testing the stated hypotheses?

Reviewer #1: Yes

Reviewer #2: Yes

3. Is the methodology feasible and described in sufficient detail to allow the work to be replicable?

Reviewer #1: Yes

Reviewer #2: Yes

4. Have the authors described where all data underlying the findings will be made available when the study is complete?

Reviewer #1: Yes

Reviewer #2: Yes

5. Is the manuscript presented in an intelligible fashion and written in standard English?

Reviewer #1: Yes

Reviewer #2: Yes

6. Review Comments to the Author

You may also provide optional suggestions and comments to authors that they might find helpful in planning their study.

Reviewer #1: Review: The effect of mother-infant group music classes on postnatal depression – a systematic review protocol

Altogether, I think this is an interesting research proposal with direct impact on clinical and societal practice. The authors propose a thorough systematic review that follows all relevant guidelines, and I believe that it is a well thought-out setup. I also appreciate the intersectional approach of the proposed analysis, where the authors propose to also look at mothers who are socially disadvantaged.

However, I do think there are some concerns that the authors should address in their study. These are mainly concerned with the inclusion criteria regarding postpartum depression and the risk of selection bias and drop-outs in the studies that are included in the review, which could skew the overall results. I believe that taking these factors into account could improve the quality and generalizability of this study.

- One of the main outcomes of the study is postnatal depression. Could the authors maybe specify which kind of outcomes they will include? Will they only include studies that conducted clinical interviews or clinically validated questionnaires, or will they also include studies that used other questions to measure postpartum depression?

- I see that the inclusion criteria also do not contain a “threshold” or criterium for postpartum depression. Do the authors propose to include all studies that look at effects of music classes on postnatal depression, even when the studies only included women without postpartum depression? Or do they plan to only include studies which included women with postpartum depression, and if so, what are the diagnosis criteria?

o A short note on the reporting of the participant demographics:

I think it is also important to report participants’ level of postpartum depression before the onset of the study, so that it is clear to which postpartum depression group the results can be generalized.

- I see that the authors propose to include all studies on group music classes, regardless of the setting in which the class is organized. This might induce a possible “interest bias”, where mothers who are actively seeking out music classes are more likely to participate and benefit from the classes. Do the authors plan on addressing a form of “interest bias”, meaning women who are more interested in the classes might join the classes?

o This form of interest bias could be accounted for through only including studies which have a control group or “regular care” group as comparison group, but the authors propose to also include studies that have no control group. Do they plan on conducting separate meta-analyses or comparisons for studies with or without a control group?

- My other concern in the included studies is the amount of time and effort it will take the participating mothers to participate in the music classes. These classes could take quite some time, maybe during the day on work days, meaning that maybe mothers with more time on their hands and mothers who feel more motivated and “up for it” might be more likely to complete the study. This could introduce a form of selective drop-out bias, where mothers who have less time or energy might be more likely to drop out.

o How do the authors plan on addressing drop-out participants per study, and the possible bias this introduces in the results?

- Motherhood and maternity leave are also very dependent on national contexts. The authors do propose to do a subgroup analysis for studies which include socially disadvantaged mothers, but do they plan on accounting for or reporting other international differences, such as national policies on maternity leave or the amount of involvement of the other parent?

o It could be interesting to report the setting in which participating mothers are in, for example, do the music classes take place in the evenings or during maternity leave so that they do not have to take time off work?

Altogether, I appreciate the rather practical approach of the proposed review. I believe that this study could be an interesting starting point to consider the feasibility of music classes for women with postpartum depression, especially considering the possible improvement of social support and maternal-infant bonding for participating women. I wish the researchers the best of luck with this interesting line of research!

Reviewer #2: I thank the Editor and authors for the opportunity to review a manuscript. The paper has overall a very good technical content and it’s easily readable. I congratulate the authors on a very interesting proposal of a systematic review with a meta-analysis. I also believe the importance of this review paper. I offer the following minor comments.

1. The authors provided information for the review in the context of what is already known. However, they wrote that “are not aware of any systematic reviews specifically addressing the effect of mother-infant group music classes on postnatal depression”. It was not clearly stated in the manuscript that a search of resources for existing or ongoing reviews was taken. I recommend to add information what kind of resources/databases have been checked to ensure the current review is justified.

2. I would recommend to consider the piloting the study selection process by applying the inclusion criteria to a sample of papers in order to check that they can be reliably interpreted and that they classify the studies appropriately.

3. Please state how extracting data from multiple reports of the same study will be done (Each report separately, then combine information across multiple data collection forms OR data from all reports directly into a single data collection form.)

4. Please expand all abbreviations used in the manuscript. eg, EPDS, GAD 7, PHQ-9 was not explained while WHO abbreviation was introduced two times.

5. I found that a searching strategy is dedicated to RCTs, and since the authors plan to include non-randomized studies as well, it might be wise to not add keywords as “Randomized Controlled Trials” or “Random Allocation”.

6. At the same time, I would be pleased if the authors could consider to restrict the eligibility criteria to RCTs only for the reliability of the data. And if so, my previous comment (no 5) would be not relevant then.

7. PLOS authors have the option to publish the peer review history of their article (what does this mean?). If published, this will include your full peer review and any attached files.

Reviewer #1: No

Reviewer #2: **Yes: **Łucja Bieleninik

---

## [Author Response · Author response to Decision Letter 0]

1 Jul 2022

Please see attached document 'Response to Reviewers' for all required actions.

---

## [Decision Letter · Decision Letter 1]

12 Aug 2022

The effect of mother-infant group music classes on postnatal depression – a systematic review protocol

PONE-D-21-35789R1

Dear Dr. Colella,

We’re pleased to inform you that your manuscript has been judged scientifically suitable for publication and will be formally accepted for publication once it meets all outstanding technical requirements.

Kind regards,

Jianhong Zhou

Staff Editor

PLOS ONE

Additional Editor Comments (optional):

Reviewers' comments:

Reviewer's Responses to Questions

**Comments to the Author**

1. Does the manuscript provide a valid rationale for the proposed study, with clearly identified and justified research questions?

Reviewer #1: Yes

Reviewer #2: Yes

2. Is the protocol technically sound and planned in a manner that will lead to a meaningful outcome and allow testing the stated hypotheses?

Reviewer #1: Yes

Reviewer #2: Yes

3. Is the methodology feasible and described in sufficient detail to allow the work to be replicable?

Reviewer #1: Yes

Reviewer #2: Yes

4. Have the authors described where all data underlying the findings will be made available when the study is complete?

Reviewer #1: Yes

Reviewer #2: Yes

5. Is the manuscript presented in an intelligible fashion and written in standard English?

Reviewer #1: Yes

Reviewer #2: Yes

6. Review Comments to the Author

You may also provide optional suggestions and comments to authors that they might find helpful in planning their study.

Reviewer #1: I think the authors provided very well-thought out and thorough answers to the reviewer comments, and their clarifications have improved the proposal strongly. I wish them the best of luck conducting this interesting review!

Reviewer #2: Dear authors,

thank you for your careful revision of your manuscript. It was a pleasure to read this revised manuscript, and I appreciate the author’s consideration of my previous feedback. This manuscript is stronger since the initial submission. All comments have been addressed satisfactorily.

Best regards!

7. PLOS authors have the option to publish the peer review history of their article (what does this mean?). If published, this will include your full peer review and any attached files.

Reviewer #1: No

Reviewer #2: No

---

## [Editor Report · Acceptance letter]

23 Aug 2022

PONE-D-21-35789R1 

The effect of mother-infant group music classes on postnatal depression – a systematic review protocol 

Dear Dr. Colella:

I'm pleased to inform you that your manuscript has been deemed suitable for publication in PLOS ONE. Congratulations! Your manuscript is now with our production department. 

Kind regards, 

on behalf of

Dr. Astrid M. Kamperman 

Academic Editor

PLOS ONE